# Simultaneous Determination and Pharmacokinetic Characterization of Glycyrrhizin, Isoliquiritigenin, Liquiritigenin, and Liquiritin in Rat Plasma Following Oral Administration of Glycyrrhizae Radix Extract

**DOI:** 10.3390/molecules24091816

**Published:** 2019-05-10

**Authors:** You Jin Han, Bitna Kang, Eun-Ju Yang, Min-Koo Choi, Im-Sook Song

**Affiliations:** 1College of Pharmacy and Research Institute of Pharmaceutical Sciences, Kyungpook National University, Daegu 41566, Korea; gksdbwls2@nate.com (Y.J.H.); ejy125@gmail.com (E.-J.Y.); 2College of Pharmacy, Dankook University, Cheon-an 31116, Korea; qlcska8520@naver.com

**Keywords:** Glycyrrhizae Radix extract, glycyrrhizin, isoliquiritigenin, liquiritigenin, liquiritin, LC–MS/MS analysis, pharmacokinetics

## Abstract

Glycyrrhizae Radix is widely used as herbal medicine and is effective against inflammation, various cancers, and digestive disorders. We aimed to develop a sensitive and simultaneous analytical method for detecting glycyrrhizin, isoliquiritigenin, liquiritigenin, and liquiritin, the four marker components of Glycyrrhizae Radix extract (GRE), in rat plasma using liquid chromatography-tandem mass spectrometry and to apply this analytical method to pharmacokinetic studies. Retention times for glycyrrhizin, isoliquiritigenin, liquiritigenin, and liquiritin were 7.8 min, 4.1 min, 3.1 min, and 2.0 min, respectively, suggesting that the four analytes were well separated without any interfering peaks around the peak elution time. The lower limit of quantitation was 2 ng/mL for glycyrrhizin and 0.2 ng/mL for isoliquiritigenin, liquiritigenin, and liquiritin; the inter- and intra-day accuracy, precision, and stability were less than 15%. Plasma concentrations of glycyrrhizin, isoliquiritigenin, liquiritigenin, and liquiritin were quantified for 24 h after a single oral administration of 1 g/kg GRE to four rats. Among the four components, plasma concentration of glycyrrhizin was the highest and exhibited a long half-life (23.1 ± 15.5 h). Interestingly, plasma concentrations of isoliquiritigenin and liquiritigenin were restored to the initial concentration at 4–10 h after the GRE administration, as evidenced by liquiritin biotransformation into isoliquiritigenin and liquiritigenin, catalyzed by fecal lysate and gut wall enzymes. In conclusion, our analytical method developed for detecting glycyrrhizin, isoliquiritigenin, liquiritigenin, and liquiritin could be successfully applied to investigate their pharmacokinetic properties in rats and would be useful for conducting further studies on the efficacy, toxicity, and biopharmaceutics of GREs and their marker components.

## 1. Introduction

Glycyrrhizae Radix (licorice root) has been used as a herbal medicine because of a variety of pharmacological activities, including anti-oxidative, anti-cancer, and anti-diabetic activities as well as memory enhancing and inflammation reducing effects [1]. In addition, it has been used as a flavoring agent in food products [2] and also as an adjuvant to increase the therapeutic efficacy of other drugs. For example, Glycyrrhizae Radix lowers the risk of aphthous ulcers caused by aspirin intake and the side effect of spironolactone. It also increased the efficacy of corticosteroids and improved the elimination of nitrofurantoin [3]. Recently, our group prepared an ethanol extract of Glycyrrhizae Radix and investigated the efficacy of Glycyrrhizae Radix extract (GRE) in relation to the modulation of reactive splenic T cells. The oral administration of GRE (0.1–0.5 g/kg) for 9 days could effectively ameliorate interferon-γ-related autoimmune responses in a mouse model of experimental autoimmune encephalomyelitis [1]. For understanding the relationship between the response elicited by GRE and its pharmacokinetics, it was important to carry out the bioanalysis of the predominant or pharmacological components of GRE in biological samples following herbal extract administration and to understand their pharmacokinetics.

Glycyrrhizae Radix contains many bioactive saponins and flavonoids along with glycyrrhizin, a major and marker component of Glycyrrhizae Radix [2,4,5,6]. It has recently been reported that glycyrrhizin exerts strong neuroprotective effects on a mouse model of experimental autoimmune encephalomyelitis [1,7]. In addition, glycyrrhizin is commonly used owing to its therapeutic effects against arthritis, hepatotoxicity, leukemia, allergies, stomach ulcers, and inflammation. Moreover, the major active flavonoids of Glycyrrhizae Radix, such as isoliquiritigenin, liquiritin, and liquiritigenin [8], are often used as anti-depressants or as anticancer, cardio-protective, anti-microbial, and neuroprotective agents [9,10,11,12,13,14]. Based on the literature search, glycyrrhizin, isoliquiritigenin, liquiritin, and liquiritigenin were selected as predominant or pharmacological components of GRE. An analytical method for simultaneously detecting these four components from a traditional Chinese herbal formulation Sijunzi decoction or from Glycyrrhizae Radix, using high-performance liquid chromatography (HPLC) with a detection limit of >300 ng/mL has been previously reported [15,16]. The previous analytical methods and pharmacokinetic studies on bioactive saponins and flavonoids following GRE administration mainly focused on pharmacokinetic drug–drug interaction between the Jiegeng and Gancao or the co-extract of Shaoyao Gancao decoction [17,18,19]. Moreover, their pharmacokinetic application has been carried out at a high dose of the extract. Plasma concentrations of the 10 active constituents including glycyrrhizin, isoliquiritigenin, liquiritin, and liquiritigenin were determined following a single oral administration of 9.5 g/kg Shaoyao–Gancao decoction extract [17]. Mao et al. determined the plasma concentrations of glycyrrhizin, glycyrrhetinic acid, isoliquiritigenin, liquiritigenin, isoliquiritin, and liquiritin after administering a single oral dose of 20 g/kg of GRE [18]. Shan et al. determined the pharmacokinetics of nine active components including these four components on repeated oral administration of Zushima–Gancao extract (2.7 g/kg) for 20 days [19]. A high dose of GRE might be administered owing to the lower concentration of active components. Even the concentration of glycyrrhizin, which was the highest in GRE, was <2% and that of other flavones was <1% [2,4,5,6].

Therefore, the purpose of this study was to establish simultaneous and sensitive assays to quantify the major and pharmacologically active components in GRE, such as glycyrrhizin, isoliquiritigenin, liquiritin, and liquiritigenin, and to implement the developed method in the pharmacokinetic studies of these four components following a single oral administration of RGE (1 g/kg) in rats. The method was developed using triple quadrupole liquid chromatography–tandem mass spectrometry (LC–MS/MS) and to validate this bioanalytical method in terms of linearity, selectivity, accuracy, precision, recovery, stability, and matrix effects according to the U.S. Food and Drug Administration Guideline for Bioanalytical Method [20].

## 2. Results

### 2.1. LC–MS/MS Analysis

#### 2.1.1. MS/MS Analysis

In order to optimize ESI conditions for four components, each compound was injected directly into the mass spectrometer ionization source. Glycyrrhizin showed optimal ionization in positive mode and isoliquiritigenin, liquiritigenin, and liquiritin showed optimal ionization in negative mode. Figure 1 shows the mass spectra and chemical structure of glycyrrhizin, isoliquiritigenin, liquiritigenin, and liquiritin. The selection of berberine as an internal standard (IS) was based on its simultaneous determination with glycyrrhizin, which was present at the highest concentration in GRE, in a positive ionization mode [21,22]. In addition, berberine eluted in the middle of glycyrrhizin and the three flavones and it showed a stable extraction recovery with low coefficient of variation (CV). The optimized analytical conditions including mass transition from the precursor to product ion (m/z) for glycyrrhizin, isoliquiritigenin, liquiritigenin, liquiritin, and berberine (IS) are listed in Table 1.

#### 2.1.2. Specificity

Representative multiple reaction monitoring (MRM) chromatograms of glycyrrhizin, isoliquiritigenin, liquiritigenin, and liquiritin, and IS (Figure 2) showed that the four analytes and berberine (IS) peaks were well-separated with no interfering peaks at their respective retention times corresponding to the concentration of the lower limit of quantification (LLOQ). The retention times of glycyrrhizin, isoliquiritigenin, liquiritigenin, and liquiritin, and IS were 7.8 min, 4.1 min, 3.1 min, 2.0 min, and 4.8 min, respectively, and the total run time was 10.0 min. The selectivity of the analytes was confirmed from six different blank rat plasma and plasma samples obtained from rats at 2 h after oral administration of GRE (1 g/kg) (Figure 3).

#### 2.1.3. Linearity and LLOQs

To access linearity, the calibration curve consisting of seven different concentrations of glycyrrhizin, isoliquiritigenin, liquiritigenin, and liquiritin was analyzed, and the calibration curves and equations for each component have been shown in Figure 4 and Table 2. The LLOQs for glycyrrhizin, isoliquiritigenin, liquiritigenin, and liquiritin in our analytical system was defined by a signal-to-noise ratio of >5, the precision was ≤15%, and the accuracy was 80–120%; these results have been listed in Table 2.

#### 2.1.4. Accuracy and Precision

The inter- and intra-day accuracy and precision were assessed using three concentrations of quality control (QC) samples consisting of the mixture of four analytes (Table 3). The results showed that inter- and intra-day precision for glycyrrhizin, isoliquiritigenin, liquiritin, and liquiritigenin was below 13.6%. The inter- and intra-day accuracy for glycyrrhizin, isoliquiritigenin, liquiritin, and liquiritigenin ranged from 87.4% to 112.2% (Table 3).

#### 2.1.5. Matrix Effect and Recovery

The extraction recoveries of glycyrrhizin, isoliquiritigenin, liquiritin, and liquiritigenin in the low, medium, high QC samples ranged from 70.3% to 99.1% with CV of <14.0%. The matrix effects ranged from 76.2 to 114.2% with CV of <14.8%. These results indicate that no significant interference occurred during the ionization and methanol precipitation process. The extraction recovery and matrix effect of the IS at 0.1 ng/mL were 86.2% and 108.2%, respectively (Table 4).

#### 2.1.6. Stability

It was found that the precision and accuracy of QC samples consisting of a mixture of the four analytes were within 12.9% for short-term stability, below 6.4% for post-preparative stability, and below 11.1% for three freeze–thaw cycle stability (Table 5). Therefore, glycyrrhizin, isoliquiritigenin, liquiritin, and liquiritigenin in plasma samples were found to exhibit no problems in these three stability tests during the bioanalytical procedure.

### 2.2. Contents of Glycyrrhizin, Liquiritin, Isoliquiritigenin, and Liquiritigenin in GRE

The concentration of glycyrrhizin, isoliquiritigenin, liquiritigenin, and liquiritin in GRE are summarized in Table 6. The concentration of glycyrrhizin was the highest (1.3%), consistent with the findings of previous studies [2,4,5,6]. Isoliquiritigenin, liquiritigenin, and liquiritin were present at lower concentrations (0.014%, 0.027%, and 0.38%, respectively) in GRE.

### 2.3. Plasma Concentration of Glycyrrhizin, Liquiritin, Isoliquiritigenin, and Liquiritigenin 

Next, we investigated the plasma concentrations of glycyrrhizin, isoliquiritigenin, liquiritigenin, and liquiritin following a single oral GRE administration at a dose of 1 g/kg. Plasma concentrations of glycyrrhizin, isoliquiritigenin, liquiritigenin, and liquiritin over time and their PK parameters are shown in Figure 5 and Table 7, respectively. Among the four major components present in rat plasma, glycyrrhizin was found to be maintained at the highest concentration for a period of 24 h. In contrast, the plasma concentration of liquiritin gradually reduced at an elimination half-life (T_1/2_) of 3.7 ± 2.2 h. Concentrations of isoliquiritigenin and liquiritigenin were similar and increased up to 8 h and then gradually reduced; therefore, T_max_ and MRT values of both these compounds were very similar. The area under the plasma concentration–time curve (AUC) of isoliquiritigenin and liquiritigenin were comparable but higher than that of liquiritin, despite their concentration (0.014% and 0.027%, respectively) in GRE being lower than that of liquiritin (0.38%). Collectively, these results suggested that isoliquiritigenin and liquiritigenin were transformed during the intestinal absorption process.

### 2.4. Biotransformation in the Rat Intestine

We investigated whether the biotransformation of isoliquiritigenin and liquiritigenin from liquiritin could occur in the rat intestine. Because the plasma concentration of isoliquiritigenin and liquiritigenin was increased at 4–10 h after oral GRE administration, we measured biotransformation in the rat ileum segment. After a 2 h incubation of a single component of GRE such as isoliquiritigenin, liquiritigenin, and liquiritin with rat ileum segments and intestinal contents, liquiritin was found to transform into isoliquiritigenin and liquiritigenin and the formation rate of both these compounds were similar to each other (Figure 6A–C). Moreover, although isoliquiritigenin and liquiritigenin were interchangeable, they did not transform into liquiritin (Figure 6D). 

## 3. Discussion

In this study, the newly developed analytical method for glycyrrhizin, isoliquiritigenin, liquiritigenin, and liquiritin using an LC–MS/MS system showed relatively higher sensitivity (i.e., LLOQ 2 ng/mL for glycyrrhizin and 0.2 ng/mL for isoliquiritigenin, liquiritigenin, and liquiritin) despite using lower plasma sample volume (50 μL). For example, Wang et al. implemented a protein-precipitation method and sample preparations via evaporation and reconstitution for detecting 10 active constituents in Shaoyao–Gancao decoction. The LLOQs for glycyrrhizin and three flavone compounds were 5 and 0.5 ng/mL, respectively [17]. Mao et al. applied analytical methods for glycyrrhizin, glycyrrhetinic acid, isoliquiritigenin, liquiritigenin, isoliquiritin, and liquiritin using an LC–MS/MS system with 10 and 0.4 ng/mL of LLOQ for glycyrrhizin and three flavones, respectively [18]. Additionally, previously established methods by Shan et al. applied liquid–liquid extraction which requires acidification with HCl for the extraction of glycyrrhizin and glycyrrhetinic acid and a larger plasma sample volume (100 μL) and the LLOQs for glycyrrhizin and three flavones were 1 and 0.34–0.5 ng/mL, respectively [19]. Herein, we used a protein-precipitation method with methanol containing IS rather than a previously described liquid–liquid extraction method or sample preparation via evaporation and reconstitution method [8,17,19], and then directly injected an aliquot of the supernatant after centrifugation of protein-precipitated plasma samples. 

We further validated our simple, sensitive, and simultaneous analytical method by performing a pharmacokinetic study after orally administering rats with 1 g/kg of GRE. We successfully measured the plasma concentrations of glycyrrhizin, isoliquiritigenin, liquiritin, and liquiritigenin for 24 h. However, we should take into account the fact that the pharmacological efficacy was investigated following repeated oral administration for 9 days at a dose range of 0.1–0.5 g/kg. Thus, the pharmacokinetic study involving repeated administration of GRE at a lower dose range of 0.1–0.5 g/kg needs to be performed to understand the pharmacokinetic–pharmacodynamic correlation of GRE.

The pharmacokinetic features of glycyrrhizin, including its long half-life, were consistent with those reported in previous studies [23,24]. The plasma concentrations of isoliquiritigenin and liquiritigenin were similar and showed fast elimination up to 4 h but rebounded to the initial plasma concentration over 10 h. As demonstrated in this study (Figure 6), liquiritin was hydrolyzed to isoliquiritigenin and liquiritigenin, and both isoliquiritigenin and liquiritigenin were interchangeable. The results suggested that liquiritin in GRE could be a precursor for isoliquiritigenin and liquiritigenin and is a more potent pharmacological component [25]; therefore, the hydrolysis of liquiritin could be attributed to an increased plasma concentration of isoliquiritigenin and liquiritigenin during the absorption period (from 4 to 10 h in this study). The results were consistent with the previous report that isoliquiritigenin and liquiritigenin, which were generated from liquiritin, were absorbed from the jejunum to colon with the help of gut wall enzymes and intestinal flora [26]. 

Mao et al. reported that the metabolism of some active components of Gancao, including glycyrrhizin and liquiritigenin, in rat fecal lysate was changed after co-administration with Jiegeng; consequently, this could be an important factor for alterations in the pharmacokinetic profiles of glycyrrhizin and liquiritigenin [18]. According to their results, not only Jiegeng but also Gancao changed the hydrolysis of liquiritigenin [18], which is similar to this study and could be a factor responsible for higher plasma concentration of liquiritigenin after GRE administration. In addition to this biotransformation, liquiritin permeability was much greater when added as a part of GRE as compared with the addition of an equal amount of liquiritin alone [18]. These results should also be clarified by performing a pharmacokinetic comparison between GRE administration and the administration of an equal amount of a single component as well as a comparison between intestinal absorption and metabolism; these topics are currently being investigated by our research group.

## 4. Materials and Methods

### 4.1. Materials 

Glycyrrhizin (Dipotassium Glycyrrhizinate, purity > 75.0%, for High Performance Liquid Chromatography grade) was purchased from Santa Cruz Biotechnology, Inc. (Dallas, TX, USA). Berberine chloride (internal standard, purity ≥ 98.0%), isoliquiritigenin (purity > 99.0%), liquiritin (purity > 98.0%), and liquiritigenin (purity > 97.0%) were gained from Sigma-Aldrich (St. Louis, MO, USA). Water, methanol, and other solvents were obtained from J.T. Baker Korea (Seoul, Korea) and TEDIA (Fairfield, OH, USA). All other chemicals and solvents are of reagent and analytical grade.

Glycyrrhizae Radix extract (GRE; #KNUNPM GR-2015-001, deposited at the laboratory of Natural Products Medicine in Kyungpook National University) was used [1]. Briefly, dried Glycyrrhizae Radix, imported from China to Republic of Korea in 2015, was extracted with 94% ethanol for 3 h. The extracted solution was filtrated and concentrated by a rotary evaporator to obtain the GRE.

### 4.2. Animals

Male Sprague–Dawley rats (7 weeks of age, 249 ± 6 g), obtained from Samtako Bio Korea (Osan, Kyunggido, Korea) were used for pharmacokinetic experiments. All animal procedures were approved by the Animal Care and Use Committee of Kyungpook National University (approval No. KNU-2017-0126) and conducted in accordance with the National Institutes of Health guidance for the care and the use of laboratory animals. Rats were maintained in an animal facility at the College of Pharmacy, Kyungpook National University at a temperature of 21–27 °C with 13 h light (08:00–21:00) and a relative humidity of 60 ± 5%.

### 4.3. Preparation of Calibration Curve and Quality Control Samples

Calibration curve samples containing glycyrrhizin (2 to 500 ng/mL), liquiritin (0.2 to 100 ng/mL), isoliquiritigenin (0.2 to 50 ng/mL), and liquiritigenin (0.2 to 50 ng/mL) were prepared using an internal standard method. Briefly, aliquots of calibration curve samples (50 µL) was added to 300 µL of methanol containing 0.1 ng/mL berberine, vortex-mixed for 10 min, and centrifuged at 10,000× *g* for 10 min at 4 °C. An aliquot (10 µL) of the supernatant was injected into the LC–MS/MS system.

### 4.4. LC–MS/MS Analysis of Glycyrrhizin, Liquiritin, Isoliquiritigenin, and Liquiritigenin

#### 4.4.1. LC–MS/MS Condition

The LC–MS/MS system connected to Agilent 6470 triple-quadrupole mass spectrometer (Agilent, Wilmington, DE, USA) via electrospray ionization (ESI) interface. The mobile phase consisted of methanol: water (65:35, *v:v*) with 0.1% formic acid. The analytical column of a Synergi Polar-RP (4 µm particle size, 150 × 2 mm, Phenomenex, Torrance, CA, USA) equipped with an guard column (4 × 2 mm, Phenomenex) was used. The sample injection volume, dwell time, and flow rate were 10 µL, 10 min, and 0.3 mL/min, respectively.

#### 4.4.2. Specificity

Six individual blank plasma samples from rats were used and specificity was defined as no interfering signal at the peak region of each analyte (glycyrrhizin, isoliquiritigenin, liquiritigenin, and liquiritin) and internal standard from background and endogenous signal.

#### 4.4.3. Linearity

The linearity of the method was evaluated by analyzing a series of calibration samples consisting glycyrrhizin (from 2 to 500 ng/mL), liquiritin (from 0.2 to 100 ng/mL), isoliquiritigenin and liquiritigenin (from 0.2 to 50 ng/mL). Least square linear regression equation of the peak area ratios of each analyte to IS against the corresponding concentrations was obtained with a weighting factor of 1/x^2^.

#### 4.4.4. Accuracy and Precision

The accuracy and precision of intra-day and inter-day were analyzed by six replicate QC samples on the same day and six consecutive days, respectively.

Accuracy is described in relative percentage of measured concentration compared to the spiked concentration and precision are determined by relative standard deviation compared to the average concentration of QC samples.

#### 4.4.5. Extraction Recovery and Matrix Effect

Extraction recovery was calculated by comparing the peak areas of each analytes in QC samples through the extraction process with those in blank plasma extracts spiked with corresponding concentrations [27]. Matrix effects were monitored by dividing the peak areas in blank plasma extracts spiked with QC concentrations by those in neat solutions of corresponding concentrations [27].

#### 4.4.6. Stability

The stability of four analytes in the rat plasma was tested from QC samples exposed to three different conditions [28]. Short-term stability was calculated by comparing QC samples that were stored for 5 h at 25 °C before sample preparation with the untreated QC samples. The three freeze–thaw cycle stability was analyzed by comparing QC samples that underwent three freeze–thaw cycles (−80 °C to 25 °C and standing for 3 h at 25 °C defined as one cycle) with the untreated QC samples. Post-preparative stability was evaluated by comparing the post-preparative QC samples maintained in the autosampler at 4 °C for 12 h with the untreated QC samples [28].

### 4.5. Determination of Glycyrrhizin, Isoliquiritigenin, Liquiritigenin, and Liquiritin in GRE

One hundred mg of GRE was diluted 50 times with methanol, and the diluted samples (50 μL) were added to 300 µL of methanol containing berberine (0.1 ng/mL), vortex-mixed for 10 min, and centrifuged at 10,000 × *g* for 10 min at 4 °C. An aliquot (10 µL) of the supernatant was injected into the LC–MS/MS system.

### 4.6. Pharmacokinetic Study 

Rats were fasted for at least 12 h with water ad libitum before pharmacokinetic experiments and femoral arteries of rats were cannulated with polyethylene tube (PE50, Jungdo, Seoul, Korea) under anesthesia with isoflurane (30 mmol/kg). 

GRE (1 g/kg, 2 mL/kg suspended in distilled water) was administered orally to rats by oral gavage. Blood samples (about 250 μL) were taken via the femoral artery at 0, 0.25, 0.5, 0.75, 1, 2, 4, 6, 8, 10 and 12 h and centrifuged at 10,000× *g* for 10 min at 4 °C. Obtained plasma (50 μL) were stored at −80 ˚C until analysis.

Glycyrrhizin, isoliquiritigenin, liquiritigenin, and liquiritin concentrations in plasma samples were analyzed using the developed LC–MS/MS method. Plasma samples (50 µL) were added to 300 µL of methanol containing berberine (0.1 ng/mL), vortex-mixed for 10 min, and centrifuged at 10,000× *g* for 10 min at 4 °C. An aliquot (10 µL) of the supernatant was injected into the LC–MS/MS system.

Pharmacokinetic parameters of glycyrrhizin, isoliquiritigenin, liquiritigenin, and liquiritin were calculated from the plasma concentration vs. time profile using the WinNonlin software (version 5.0, Certara Inc., Princeton, NJ, USA).

### 4.7. Biotransformation of Isoliquiritigenin and Liquiritigenin from Liquiritin

Rats were fasted for at least 12 h with water ad libitum before performing dissection of the rat ileum. The ileal segment (approximately 20 cm) was excised after the rats were euthanized by cervical dislocation. The dissected ileal segments were washed using a 10 mL syringe filled with 30 mL pre-warmed Hank’s balanced salt solution (HBSS, pH 7.4; Sigma, St. Louis, MO, USA); the eluent was vortexed for 1 min followed by centrifugation at 1000× *g* for 5 min at 4 °C. The supernatant was used for incubation with isoliquiritigenin, liquiritigenin, and liquiritin. The ileal segments were mounted onto the Ussing chambers (Navicyte, Holliston, MA, USA) and were acclimatized with HBSS for 30 min. 

To mimic the intestinal situation that occurred when GRE was administered to rats via the oral route, the concentrations of isoliquiritigenin, liquiritigenin, and liquiritin were determined based on the oral dose of GRE, their content in GRE, and fluid volume of the small intestine. For example, 1 g of GRE suspended in 2 mL water was administered at a dose of 1 g/kg to rats having stomach fluid volumes of 3.2–7.8 mL [29], which resulted in a 5–10-fold dilution. Therefore, 14 μg/mL of isoliquiritigenin, 27 μg/mL of liquiritigenin, or 380 μg/mL of liquiritin was present in the intestinal diluent.

The experiments began by changing HBSS with pre-warmed intestinal eluent (1 mL) containing 14 μg/mL of isoliquiritigenin, 27 μg/mL of liquiritigenin, or 380 μg/mL of liquiritin to the apical side of the ileal segment, followed by 2 h of incubation. Carbogen gas (5% CO_2_/ 95% O_2_) was bubbled into the Ussing chambers at a rate of 150 drops/min during the experiment. Aliquots (50 μL) of the samples were mixed with 300 µL of methanol containing berberine (0.1 ng/mL), vortex-mixed for 10 min, and centrifuged at 10,000× *g* for 10 min at 4 °C. An aliquot (10 µL) of the supernatant was injected into the LC–MS/MS system.

## 5. Conclusions

A sensitive and simultaneous LC–MS/MS method for the determination of the four marker components of Glycyrrhizae Radix, namely glycyrrhizin, isoliquiritigenin, liquiritin, and liquiritigenin, has been developed and validated in rat plasma; this analytical method was successfully applied to investigate the pharmacokinetic profiles of glycyrrhizin, isoliquiritigenin, liquiritin, and liquiritigenin in rats following a single oral administration of GRE (1 g/kg) for 24 h. 

This method can easily be applied in the bioanalysis and pharmacokinetic studies of GRE, including its administration at multiple therapeutic doses, or for making pharmacokinetic comparisons among individual components in small experimental animals. Moreover, following an appropriate validation, the present method can be extended to determine routine drug monitoring of glycyrrhizin, isoliquiritigenin, liquiritin, and liquiritigenin in plasma samples as well as in other biological samples, and thus, can be applied to in vivo pharmacokinetic–pharmacodynamic correlation studies.

## Figures and Tables

**Figure 1 molecules-24-01816-f001:**
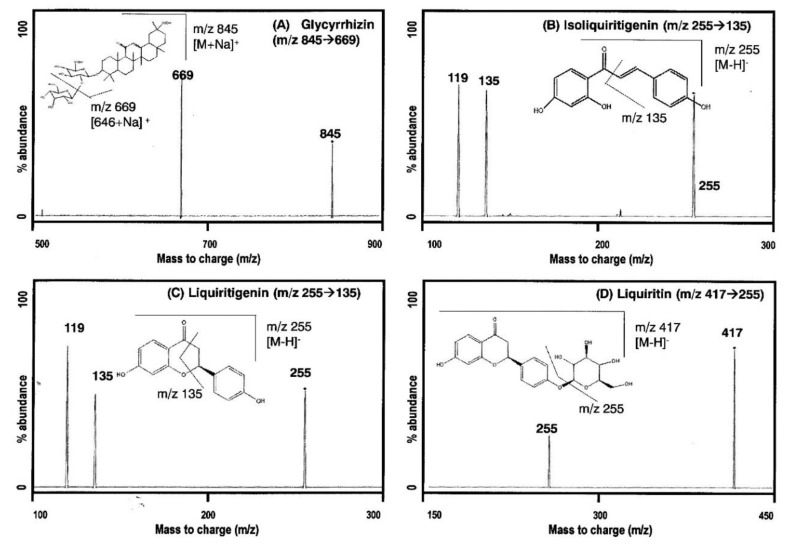
Structure and product ion mass spectra of (**A**) glycyrrhizin, (**B**) isoliquiritigenin, (**C**) liquiritigenin, and (**D**) liquiritin.

**Figure 2 molecules-24-01816-f002:**
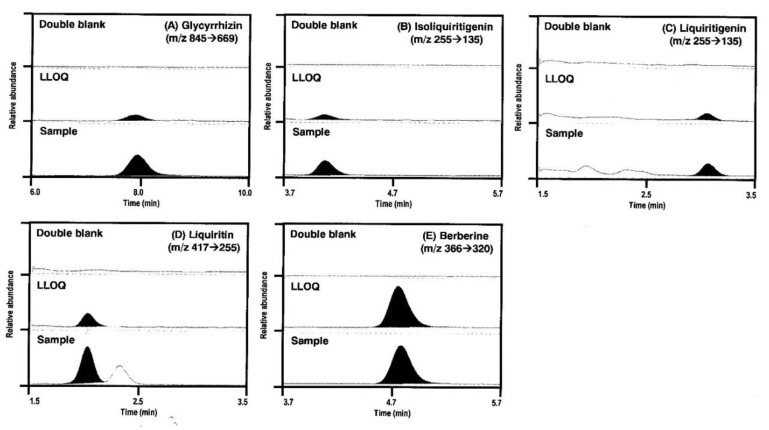
Representative multiple reaction monitoring (MRM) chromatograms of (**A**) glycyrrhizin, (**B**) isoliquiritigenin, (**C**) liquiritigenin, (**D**) liquiritin, and (**E**) berberine (IS) in rat double blank plasma (upper), rat blank plasma spiked with standard solution at lower limit of quantification (LLOQ) (center), and rat plasma samples at 2 h following single oral administration of Glycyrrhizae Radix extract (GRE) (1 g/kg) (lower).

**Figure 3 molecules-24-01816-f003:**
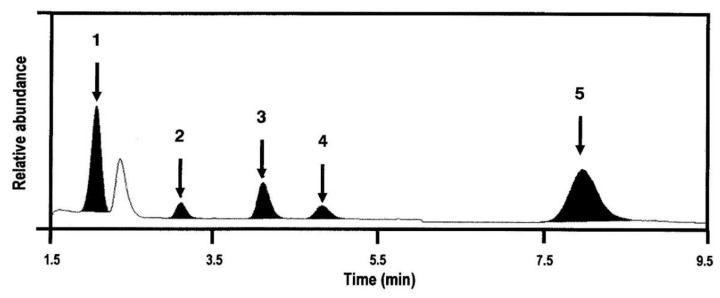
Representative MRM chromatograms of glycyrrhizin (**5**), isoliquiritigenin (**3**), liquiritigenin (**2**), liquiritin (**1**), and berberine (**4**) in rat plasma samples at 2 h following single oral administration of GRE (1 g/kg).

**Figure 4 molecules-24-01816-f004:**
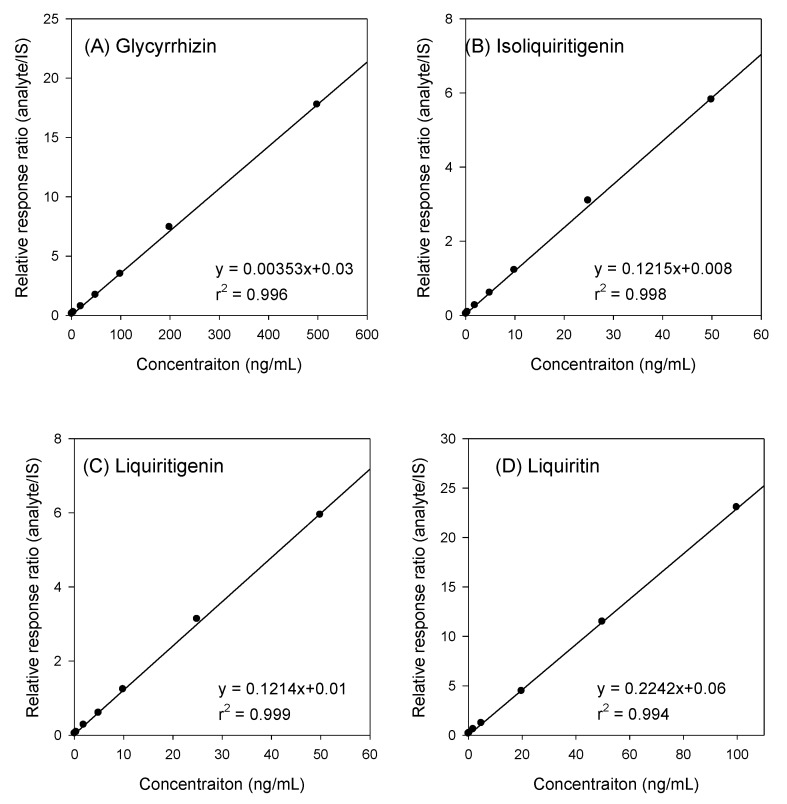
Representative calibration curves of (**A**) glycyrrhizin, (**B**) isoliquiritigenin, (**C**) liquiritigenin, and (**D**) liquiritin in rat plasma.

**Figure 5 molecules-24-01816-f005:**
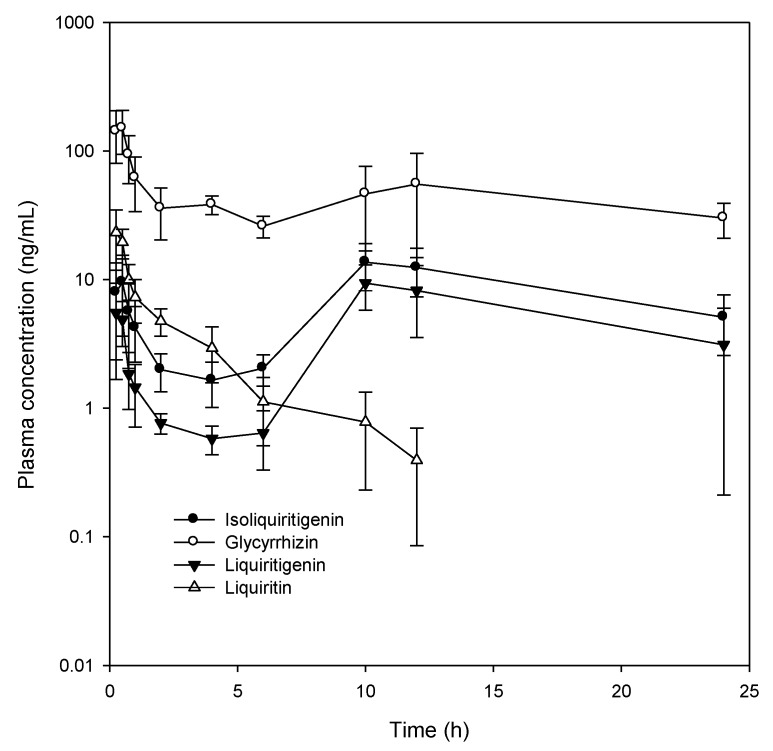
Plasma concentration–time profiles of glycyrrhizin, isoliquiritigenin, liquiritigenin, and liquiritin and after oral administration of GRE to rats. Data were expressed as mean ± SD from four rats per group.

**Figure 6 molecules-24-01816-f006:**
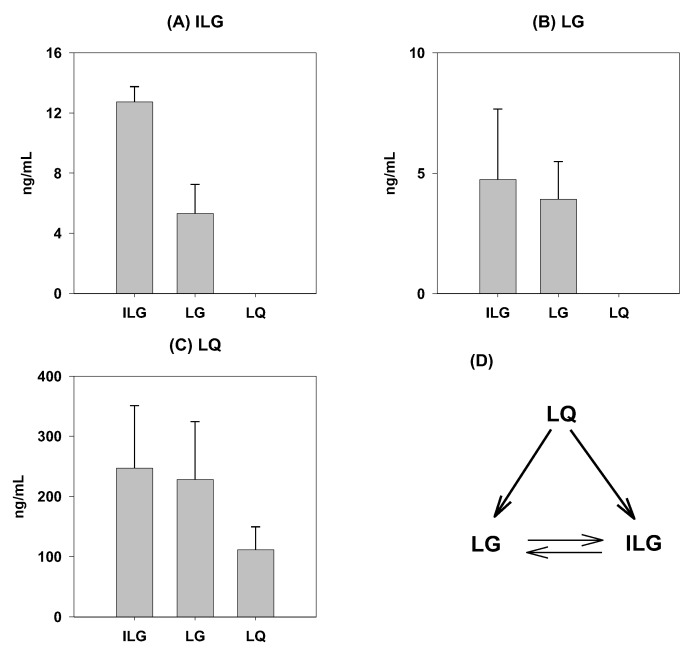
Formation of isoliquiritigenin (ILG), liquiritigenin (LG), and liquiritin (LQ) 2 h incubation after the addition of (**A**) isoliquiritigenin (ILG), (**B**) liquiritigenin (LG), and (**C**) liquiritin (LQ) in rat ileum. (**D**) Proposed biotransformation scheme among LQ, LG, and ILG in rat intestine. Data were expressed as mean ± SD from six independent experiments per group.

**Table 1 molecules-24-01816-t001:** MS/MS parameters for the detection of the analytes and IS.

Compounds	MRM Transitions (m/z)		Ionization Mode	Collision Energy (eV)
	Precursor Ion	Product Ion		
Glycyrrhizin	845	669	Positive	35
Isoliquiritigenin	255	135	Negative	15
Liquiritigenin	255	135	Negative	15
Liquiritin	417	255	Negative	20
Berberine (IS)	336	320	Positive	30

**Table 2 molecules-24-01816-t002:** Linearity and LLOQs of glycyrrhizin, isoliquiritigenin, liquiritin and liquiritigenin.

Analytes	Representative Regression Equation	r^2^	Linear Range (ng/mL)	LLOQ (ng/mL)
Glycyrrhizin	y = 0.0035x + 0.03	0.996	2–500	2
Isoliquiritigenin	y = 0.1215x + 0.008	0.998	0.2–50	0.2
Liquiritigenin	y = 0.1214x + 0.01	0.999	0.2–50	0.2
Liquiritin	y = 0.2242x + 0.06	0.994	0.2–100	0.2

**Table 3 molecules-24-01816-t003:** Intra- and inter-day precision and accuracy of glycyrrhizin, isoliquiritigenin, liquiritin and liquiritigenin.

Analytes	Nominal Concentration (ng/mL)	Intra-day			Inter-day		
		Measured Concentration (ng/mL)	Precision (%)	Accuracy (%)	Measured Concentration (ng/mL)	Precision (%)	Accuracy (%)
Glycyrrhizin	6	5.8	13.3	96.1	5.2	6.3	87.4
	75	70.1	13.6	93.5	77.5	6.8	103.3
	400	416.8	13.3	104.2	410.1	7.4	102.5
Isoliquiritigenin	0.6	0.7	4.9	108.6	0.6	8.9	101.8
	7.5	7.9	10.8	105.6	7.7	4.7	102.1
	30	32.8	7.9	109.2	33.3	7.0	111.0
Liquiritigenin	0.6	0.6	4.0	98.6	0.6	7.6	97.7
	7.5	7.7	3.7	103.0	7.5	5.8	100.5
	30	32.6	3.7	108.6	33.6	5.6	112.0
Liquiritin	0.6	0.6	9.8	99.2	0.6	8.8	103.3
	10	9.9	10.0	99.2	10.0	5.5	99.9
	75	74.2	8.9	98.9	84.1	3.8	112.2

Data represented as mean ± SD from six independent experiments.

**Table 4 molecules-24-01816-t004:** Extraction recoveries and matric effects for the determination of liquiritin, liquiritigenin, isoliquiritigenin, glycyrrhizin and of IS.

Analyte		Concentration (ng/mL)	Extraction Recovery (%)	CV (%)	Matrix Effects (%)	CV (%)
Glycyrrhizin	Low QC	6	89.06 ± 7.38	8.29	98.80 ± 5.89	5.96
	Medium QC	75	77.2 ± 7.7	9.9	92.1 ± 3.6	3.9
	High QC	400	75.0 ± 4.9	6.6	96.4 ± 4.6	4.8
Isoliquiritigenin	Low QC	0.6	80.2 ± 10.8	13.6	78.9 ± 3.0	3.8
	Medium QC	7.5	74.6 ± 8.1	10.8	88.3 ± 4.9	5.5
	High QC	30	70.3 ± 4.9	7.1	93.6 ± 5.3	5.6
Liquiritigenin	Low QC	0.6	99.1 ± 7.9	8.0	76.2 ± 3.1	4.1
	Medium QC	7.5	88.9 ± 9.5	10.7	96.6 ± 8.1	8.4
	High QC	30	83.5 ± 6.4	7.7	104.9 ± 8.2	7.8
Liquiritin	Low QC	0.6	79.2 ± 11.1	14.0	114.2 ± 10.4	9.1
	Medium QC	10	83.2 ± 11.3	13.6	97.2 ± 14.4	14.8
	High QC	75	90.7 ± 6.4	7.1	101.8 ± 7.7	7.6
IS		0.1	86.2 ± 2.7	3.1	108.2 ± 1.8	1.7

Data represented as mean ± SD from six independent experiments.

**Table 5 molecules-24-01816-t005:** Stability of glycyrrhizin, isoliquiritigenin, liquiritin and liquiritigenin.

Storage Conditions	Analytes	Concentration (ng/mL)		Precision %	Accuracy %
		Spiked	Measured		
Short-term stability	Glycyrrhizin	6	5.6	6.8	94.0
		400	390.8	9.5	97.7
	Isoliquiritigenin	0.6	0.6	12.9	101.8
		30	30.1	7.5	100.3
	Liquiritigenin	0.6	0.6	3.3	93.6
		30	28.8	8.9	102.7
	Liquiritin	0.6	0.6	9.1	92.8
		75	71.3	10.1	99.5
Post-preparative stability	Glycyrrhizin	6	5.8	4.9	96.2
		400	390.8	3.7	97.7
	Isoliquiritigenin	0.6	0.6	4.1	92.4
		30	32.2	3.6	107.3
	Liquiritigenin	0.6	0.6	6.4	93.0
		30	32.8	4.8	109.5
	Liquiritin	0.6	0.6	3.7	97.3
		75	76.4	2.5	101.9
Three freeze-thaw cycle stability	Glycyrrhizin	6	6.8	0.8	113.9
		400	394.1	10.3	98.5
	Isoliquiritigenin	0.6	0.5	5.8	90.9
		30	30.0	7.1	99.9
	Liquiritigenin	0.6	0.5	5.1	90.0
		30	29.2	8.2	97.2
	Liquiritin	0.6	0.5	4.3	87.3
		75	72.6	11.1	96.7

Data represented as mean ± SD from six independent experiments.

**Table 6 molecules-24-01816-t006:** The mean contents of four compounds in GRE.

Compounds	Content (%)
Glycyrrhizin	1.3 ± 0.2
Isoliquiritigenin	0.014 ± 0.004
Liquiritigenin	0.027 ± 0.010
Liquiritin	0.38 ± 0.07

Data represented as mean ± SD from six independent experiments.

**Table 7 molecules-24-01816-t007:** Pharmacokinetic parameters of glycyrrhizin, isoliquiritigenin, liquiritin, liquiritigenin after oral administration of GRE (1 g/kg) to rats.

Parameters	Glycyrrhizin	Isoliquiritigenin	Liquiritigenin	Liquiritin
C_max_ (ng/mL)	164.4 ± 62.0	16.6 ± 2.7	10.8 ± 3.6	26.8 ± 8.5
AUC_last_ (ng∙h/mL)	1051.0 ± 487.5	179.2 ± 46.3	112.5 ± 36.4	39.5 ± 7.8
T_max_ (h)	0.4 ± 0.1	8.1 ± 5.2	8.1 ± 5.3	0.4 ± 0.1
T_1/2_ (h)	23.1 ± 15.5	-	-	3.7 ± 2.2
MRT_last_ (h)	10.7 ± 0.7	12.5 ± 1.3	12.8 ± 1.8	3.3 ± 1.3

Data were expressed as mean ± SD from four rats; C_max_: maximum plasma concentration; AUC_last_: Area under plasma concentration-time curve from zero to last time; T_max_, time to reach C_max_; T_1/2_: elimination half-life; MRT: mean residence time.

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
