# Peer review of "Simultaneous Determination and Pharmacokinetic Characterization of Glycyrrhizin, Isoliquiritigenin, Liquiritigenin, and Liquiritin in Rat Plasma Following Oral Administration of Glycyrrhizae Radix Extract"

_molecules, 2019, doi:10.3390/molecules24091816_

Round 1

Reviewer 1 Report

the paper describe LC MSMS method for pharmacokinetic study in rat of Glycirrhizae extract.

The topic fit the journal's aims and scopes

The novelty of the findings is clearly explained but a more precise literature search on similar methods should be performed and comparation of the obtained results with other previously published methods on glycirrizae active compounds can be done.

Introduction is not clear and need careful revision.

Criteria for the selection of IS should be added.

Calibration curve is missing

chormatograms of samples as exaample can be added.

What about the use of the method when a lower dose of extract is administered? very high dose is used what is the rational for this?

Validation is well described and pharmacokinetic also.

What about the comparison of single compounds with the extract in order to assess possibile sinergic effects in absorption?

Biotransformation part is interesting and is novel and need more detail and more comments.

english need careful revision

Author Response

Response to Reviewer’s Comments

Reviewer #1:

Q1. The novelty of the findings is clearly explained but a more precise literature search on similar methods should be performed and comparation of the obtained results with other previously published methods on glycirrizae active compounds can be done.

Answer> According to the reviewer’s comment, we performed the literature search and compared these analytical methods with ours and discussed the results in the Discussion as follows:

Page 10 line 211 : In this study, the newly developed analytical method for glycyrrhizin, isoliquiritigenin, liquiritigenin, and liquiritin using an LC–MS/MS system showed relatively higher sensitivity (i.e. LLOQ 2 ng/mL for glycyrrhizin and 0.2 ng/mL for isoliquiritigenin, liquiritigenin, and liquiritin) despite using lower plasma sample volume (50 μL). For example, Wang et al. implemented a protein-precipitation method and sample preparations via evaporation and reconstitution for detecting 10 active constituents in Shaoyao–Gancao decoction. The LLOQs for glycyrrhizin and three flavone compounds were 5 and 0.5 ng/mL, respectively [17]. Mao et al. applied analytical methods for glycyrrhizin, glycyrrhetinic acid, isoliquiritigenin, liquiritigenin, isoliquiritin, and liquiritin using an LC–MS/MS system with 10 and 0.4 ng/mL of LLOQ for glycyrrhizin and 3 flavones, respectively [18]. Additionally, previously established methods by Shan et al. applied liquid–liquid extraction which requires acidification with HCl for the extraction of glycyrrhizin and glycyrrhetinic acid and a larger plasma sample volume (100 μL) and the LLOQs for glycyrrhizin and three flavones were 1 and 0.34–0.5 ng/mL, respectively [19]. Herein, we used a protein-precipitation method with methanol containing IS rather than a previously described liquid–liquid extraction method or sample preparation via evaporation and reconstitution method [8,17,19], and then directly injected an aliquot of the supernatant after centrifugation of protein-precipitated plasma samples.

Q2. Introduction is not clear and need careful revision.

Answer> We revised the Introduction part according to the reviewer’s comment as follows:

Page 2 line 46 : Recently, our group prepared an ethanol extract of Glycyrrhizae Radix and investigated the efficacy of Glycyrrhizae Radix extract (GRE) in relation to the modulation of reactive splenic T cells. The oral administration of GRE (0.1–0.5 g/kg) for 9 days could effectively ameliorate interferon-γ-related autoimmune responses in a mouse model of experimental autoimmune encephalomyelitis [1]. For understanding the relationship between the response elicited by GRE and its pharmacokinetics, it was important to carry out the bioanalysis of the predominant or pharmacological components of GRE in biological samples following herbal extract administration and to understand their pharmacokinetics.

Glycyrrhizae Radix includes many bioactive saponins and flavonoids along with glycyrrhizin, a major and marker component of Glycyrrhizae Radix [2,4-6]. It has recently been reported that glycyrrhizin exerts strong neuroprotective effects on a mouse model of experimental autoimmune encephalomyelitis [1,7]. In addition, glycyrrhizin is commonly used owing to its therapeutic effects against arthritis, hepatotoxicity, leukemia, allergies, stomach ulcers, and inflammation. Moreover, the major active flavonoids of Glycyrrhizae Radix, such as isoliquiritigenin, liquiritin, and liquiritigenin [8], are often used as anti-depressants or as anticancer, cardio-protective, anti-microbial, and neuroprotective agents [9-14]. Based on the literature search, glycyrrhizin, isoliquiritigenin, liquiritin, and liquiritigenin were selected as predominant or pharmacological components of GRE. An analytical method for simultaneously detecting these four components from a traditional Chinese herbal formulation Sijunzi decoction or from Glycyrrhizae Radix, using high-performance liquid chromatography (HPLC) with a detection limit of >300 ng/mL has been previously reported [15,16]. The previous analytical methods and pharmacokinetic studies on bioactive saponins and flavonoids following GRE administration mainly focused on pharmacokinetic drug–drug interaction between the Jiegeng and Gancao or the co-extract of Shaoyao Gancao decoction [17-19]. Moreover, their pharmacokinetic application has been carried out at a high dose of the extract. Plasma concentrations of the 10 active constituents including glycyrrhizin, isoliquiritigenin, liquiritin, and liquiritigenin were determined following a single oral administration of 9.5 g/kg Shaoyao–Gancao decoction extract [17]. Mao et al. determined the plasma concentrations of glycyrrhizin, glycyrrhetinic acid, isoliquiritigenin, liquiritigenin, isoliquiritin, and liquiritin after administering a single oral dose of 20 g/kg of GRE [18]. Shan et al. determined the pharmacokinetics of nine active components including these four components on repeated oral administration of Zushima–Gancao extract (2.7 g/kg) for 20 days [19]. A high dose of GRE might be administered owing to the lower concentration of active components. Even the concentration of glycyrrhizin, which was the highest in GRE, was <2% and that of other flavones was <1% [2,4-6].

Therefore, the purpose of this study was to establish simultaneous and sensitive assays to quantify the major and pharmacologically active components in GRE, such as glycyrrhizin, isoliquiritigenin, liquiritin, and liquiritigenin, and to implement the developed method in the pharmacokinetic studies of these four components following a single oral administration of RGE (1 g/kg) in rats.

Q3. Criteria for the selection of IS should be added.

Answer> We added the selection of berberine as an IS according to the reviewer’s comment as follows:

Page 3 line 96 : The selection of berberine as an internal standard (IS) was based on its simultaneous determination with glycyrrhizin, which was present at the highest concentration in GRE, in a positive ionization mode [21,22]. In addition, berberine eluted in the middle of glycyrrhizin and the three flavones and it showed a stable extraction recovery with low coefficient of variation (CV).

Q4. Calibration curve is missing

Answer> As the reviewer suggested, we added calibration curve in Figure 2 during the revision.

Q5. chormatograms of samples as example can be added.

Answer> As the reviewer suggested, we added representative chromatograms of glycyrrhizin, isoliquiritigenin, liquiritigenin, liquiritin, and berberine in rat plasma samples at 2 h following single oral administration of GRE (1 g/kg) in Figure 2 during the revision. Figure 3 also added to show the specificity and complete separation of 4 analytes and IS. 

Q6. What about the use of the method when a lower dose of extract is administered? very high dose is used what is the rational for this?

Answer> As we described in the Introduction, according to the previous reports, high dose of GRE (or Gancao) was administered to rats and the pharmacokinetics of representative components of GRE were investigated. In this study, we administered 1 g/kg of GRE to rats and measured the plasma concentrations of glycyrrhyzin, isoliquiritigenin, liquiritigenin, and liquiritin based on our pharmacology study, in which we orally administered GRE for 9 days at a dose range of 0.1-0.5 g/kg. Although we did not perform the pharmacokinetic study following repeated administration of GRE, we believe we can determine the four components in rat plasma following repeated administration of GRE. We also added the Discussion regarding the dose of GRE as follows:

Page 2 line 69 : Moreover, their pharmacokinetic application has been carried out at a high dose of the extract. Plasma concentrations of the 10 active constituents including glycyrrhizin, isoliquiritigenin, liquiritin, and liquiritigenin were determined following a single oral administration of 9.5 g/kg Shaoyao–Gancao decoction extract [17]. Mao et al. determined the plasma concentrations of glycyrrhizin, glycyrrhetinic acid, isoliquiritigenin, liquiritigenin, isoliquiritin, and liquiritin after administering a single oral dose of 20 g/kg of GRE [18]. Shan et al. determined the pharmacokinetics of nine active components including these four components on repeated oral administration of Zushima–Gancao extract (2.7 g/kg) for 20 days [19]. A high dose of GRE might be administered owing to the lower concentration of active components. Even the concentration of glycyrrhizin, which was the highest in GRE, was <2% and that of other flavones was <1% [2,4-6].

Page 10 line 228 : We further validated our simple, sensitive, and simultaneous analytical method by performing a pharmacokinetic study after orally administering rats with 1 g/kg of GRE. We successfully measured the plasma concentrations of glycyrrhizin, isoliquiritigenin, liquiritin, and liquiritigenin for 24 h. However, we should take into account the fact that the pharmacological efficacy was investigated following repeated oral administration for 9 days at a dose range of 0.1–0.5 g/kg. Thus, the pharmacokinetic study involving repeated administration of GRE at a lower dose range of 0.1–0.5 g/kg need to be performed to understand the pharmacokinetic–pharmacodynamic correlation of GRE.

Q7. Validation is well described and pharmacokinetic also.

Answer> Thank you for the reviewer’s positive comments.

Q8. What about the comparison of single compounds with the extract in order to assess possible sinergic effects in absorption? Biotransformation part is interesting and is novel and need more detail and more comments.

Answer> Thank you for the valuable comments. We are planning the comparison of pharmacokinetics between single component of GRE and equal amount of the extract in order to access possible interaction among the components of GRE in relation to the intestinal absorption process as well as gut wall biotransformation. We also added this issue in the Discussion. We asked generous understanding in this matter.

Page 11 line 248 : Mao et al. reported that the metabolism of some active components of Gancao, including glycyrrhizin and liquiritigenin, in rat fecal lysate was changed after co-administration with Jiegeng; consequently, this could be an important factor for alterations in the pharmacokinetic profiles of glycyrrhizin and liquiritigenin [18]. According to their results, not only Jiegeng but also Gancao changed the hydrolysis of liquiritigenin [18], which is similar to this study and could be a factor responsible for higher plasma concentration of liquiritigenin after GRE administration. In addition to this biotransformation, liquiritin permeability was much greater when added as a part of GRE as compared with the addition of an equal amount of liquiritin alone [18]. These results should also be clarified by performing a pharmacokinetic comparison between GRE administration and the administration of an equal amount of a single component as well as a comparison between intestinal absorption and metabolism; these topics are currently being investigated by our research group.

Q10. English need careful revision

Answer> According to the reviewers’ comment, we made substantial revision and the revised version of our manuscript has professional English Editing Service by native speaker (EssayReview). 

Reviewer 2 Report

The authors proposed a pharmacokinetic characterization of 4 main components of Glycyrrhizae Radix extract. The manuscript is well present, but some English check is required. I find out that the conclusion is missing therefore I only recommend for publication after some minor revisions. Figure 1 and 2 have low quality, figures with better resolutions should be provided.

Author Response

Response to Reviewer’s Comments

Reviewer #2:

Q1. The authors proposed a pharmacokinetic characterization of 4 main components of Glycyrrhizae Radix extract. The manuscript is well present, but some English check is required. I find out that the conclusion is missing therefore I only recommend for publication after some minor revisions. Figure 1 and 2 have low quality, figures with better resolutions should be provided.

Answer> Thank you for the reviewer’s positive comments. As the reviewer suggested, we added conclusion and substitute Figures 1 and 2 with high resolution (600 dpi) during the revision. And the revised version of our manuscript has professional English Editing Service by native speaker (EssayReview).

Round 2

Reviewer 1 Report

the authors replied to all my previous coments

I think that in the present fom the manuscript is acceptable